# Effects of Small-Sided Recreational Volleyball on Health Markers and Physical Fitness in Middle-Aged Men

**DOI:** 10.3390/ijerph17093021

**Published:** 2020-04-27

**Authors:** Nebojša Trajković, Goran Sporiš, Tomislav Krističević, Špela Bogataj

**Affiliations:** 1Faculty of Sport and Physical Education, University of Novi Sad, 21101 Novi Sad, Serbia; nele_trajce@yahoo.com; 2Faculty of Kinesiology, University of Zagreb, 10110 Zagreb, Croatia; goran.sporis@kif.unizg.hr (G.S.); tomislav.kristicevic@kif.unizg.hr (T.K.); 3Department of Nephrology, University Medical Centre, 1000 Ljubljana, Slovenia; 4Faculty of Sport, University of Ljubljana, 1000 Ljubljana, Slovenia

**Keywords:** team sport, intervention, impact, adults

## Abstract

The present study aimed to investigate whether recreational volleyball organized as small-sided games could improve fitness and health profiles of middle-aged men after 10 weeks of training. Twenty-four healthy men aged 35–55 were randomized in a small-sided recreational volleyball group (RV = 12; age: 44.7 ± 6.34 years; body mass index: 25.85 ± 1.74) and control group (CON = 12; age: 42.9 ± 8.72 years; body mass index: 25.62 ± 1.48). The RV group carried out a volleyball training program, whereas the CON group continued their daily life activities during this period. The participants in the RV group performed 2/3 training sessions of 90 min per week. Results from a repeated measure ANOVA indicated a significant group × time interaction for low-density lipoprotein (LDL) cholesterol (F = 6.776, *p* = 0.016, partial ƞ^2^ = 0.235) and for resting heart rate (F = 11.647, *p* = 0.002, partial ƞ^2^ = 0.346) in favor of the RV group. No significant changes were observed for body weight, body mass index, and diastolic blood pressure. Results for physical fitness indicated a significant interaction for Yo-Yo intermittent recovery test–level 2 (F = 11.648, *p* = 0.003, partial ƞ^2^ = 0.380), with no significant changes in both groups for handgrip strength. Recreational small-sided volleyball can be an effective training modality to stimulate a decrease in LDL cholesterol and resting HR with small improvements in cardiovascular fitness. Recreational volleyball played only two times per week shows positive cardiovascular fitness and health-related adaptations, which may contribute to the reduction of the risk of developing lifestyle diseases.

## 1. Introduction

Regular physical activity in adults can reduce the risk of hypertension, cardiovascular disease, osteoporosis, type 2 diabetes mellitus, obesity, anxiety, and depressiontab [1]. The American College of Sports Medicine (ACSM) recommends at least 150 min of moderate physical activity per week to see substantial health benefits in healthy adults [2]. These recommendations can also apply to people with specific disabilities or chronic diseases if a health professional tailors the activity to the individual’s physical function, exercise response, health status, and goals. Additionally, Wen et al. [3] stated that 90 min per week or 15 min per day of moderate-intensity physical exercise might benefit individuals with the risk of cardiovascular disease. However, various barriers exist that prevent adults from fulfilling this physical activity recommendation (e.g., lack of time or uncertainty about the amount of exercise required to benefit health) [3]. Nevertheless, staying physically and mentally fit is one of the essential properties in middle-aged adults’ life [4].

Volleyball represents one of the most popular sports in the world. Moreover, we know that it is an intermittent sport that requires players to compete in frequent short bouts of high-intensity exercise, followed by periods of low-intensity activity [5]. Therefore, it could be assumed that playing volleyball can develop speed, muscular power, and the ability to perform these repeated maximal efforts with limited recovery in adult healthy participants [6]. However, only limited data exist concerning the benefits of recreational volleyball in the adult population. Gouttebarge, Zwerver, and Verhagen [7] recently stated that volleyball has beneficial health effects. On the contrary, Trajkovic et al. [8] showed that recreational volleyball was not beneficial in improving cardiovascular fitness in healthy middle-aged men. However, the mentioned study was conducted on full-court six versus six players. Studies have shown that the mean heart rate values around 80% of HRmax or higher are considered sufficient to cause marked improvements in cardiorespiratory fitness, systolic blood pressure, and glucose tolerance, thus increasing the overall health profile [9,10,11]. Therefore, it would be interesting to see if small-sided recreational volleyball has better effects concerning the fact that a decrease in space and the number of players in the game allow higher intensity in the game [12]. Recreational football was recognized as an effective physical activity with many health-related benefits [13]. Moreover, recreational handball practice showed positive physical fitness and health-related adaptations, which may reduce the risk of developing lifestyle diseases [14]. Playing basketball small-sided games in recreational settings utilizes anaerobic and aerobic metabolism pathways, leading to significant improvements in musculoskeletal and cardiovascular fitness [15]. Recreational volleyball can be considered a useful, low intensity, enjoyable exercise intervention with a broad range of health and fitness benefits in former volleyball players but also for the general population. Moreover, considering the popularity of volleyball worldwide, training studies using this sport as a health and physical fitness enhancing intervention are warranted.

Although recreational 6 vs. 6 volleyball was proven to be a useful physical activity to stimulate some health benefits [8], we do not know if there are significant benefits of recreational volleyball played on a smaller court with fewer players in the middle-age population. Therefore, this study aimed to determine the effects of recreational small-sided volleyball on health markers and physical fitness in the middle-age population. We assume that improvements in participants’ physical fitness and most of the health marker variables will occur following the small-sided recreational volleyball intervention.

## 2. Materials and Methods

### 2.1. Subjects

Twenty-six healthy men aged 35–55 years took part in the study. All the participants had some previous experience with volleyball and were recruited with the help of local volleyball clubs. The participants did not take medication and had not been involved in any type of organized physical training for at least two years. The subjects were matched and randomly assigned to a small-sided recreational volleyball group (RV; *n* = 14; age: 44.7 ± 6.34 years; height: 181.8 ± 6.1 cm), or a control group—performing no organized physical training (CON; *n* = 12; age: 42.9 ± 8.72 years; height: 184.4 ± 5.4 cm). Two subjects dropped out due to injury during volleyball training (ankle sprain). The participants in RV carried out a 10-week volleyball training program, whereas the participants in CON continued their daily life activities during the period. The local Institutional Review Board approved the experimental protocol (Ref. No. 22/2018; approved in May 2018) and followed the Declaration of Helsinki.

### 2.2. Procedures

Subjects were familiarized with all test procedures. Resting heart rate and blood pressure were determined from 8 to 10 a.m. under standardized conditions after an overnight fast. Blood pressure was measured at least six times by an automatic upper arm blood pressure monitor (HEM-709; OMRON, Illinois, USA), and an average value was calculated. Body height and body weight were measured according to the instructions of the International Biological Program (IBP). Height was measured with a GPM anthropometer (Siber & Hegner, Zurich, Switzerland) to the nearest 0.1 cm. Body weight was obtained using a TANITA BC 540 (TANITA Corp., Arlington Heights, IL) to the nearest 0.1 kg. Body mass index was calculated by the formula: BMI = weight (kg) ÷ height^2^ (m^2^).

### 2.3. Blood Collection and Analysis

Blood samples were assessed before and after ten weeks of the small-sided recreational volleyball program. On the day of the testing, participants arrived at the laboratory after an overnight fast of between 8 and 10 h. A resting blood sample was taken after participants had been standing for at least 15 min. Participants abstained from alcohol and caffeine consumption for at least 24 h and did not perform any exercise 72 h before the testing.

In order to determine total plasma cholesterol, high-density lipoprotein (HDL) and low-density lipoprotein (LDL) cholesterol, triglycerides, and blood glucose, venous blood samples were collected from the antecubital arm vein (left or right) using a standard operating procedure, in the morning (8–10 a.m.) and after an overnight fast of at least 8 h. Blood markers were determined using automated analyzers, total cholesterol, HDL and LDL cholesterol, triglycerides (Siemens, Munich, Germany), and glucose (UV enzymatic assay; Automated Olympus AU5400, Beckman-Coulter equipment, Brea, USA) in a clinical laboratory.

### 2.4. Physical Fitness

The aerobic performance was assessed by the Yo-Yo intermittent recovery test–level 2 (YYIRT2) [16]. Heart rate (HR) max was considered as the highest HR value achieved from the YYIRT2 using short-range telemetry (Polar Electro Oy, Kempele, Finland).

Upper body isometric strength (handgrip strength test) was assessed using a handgrip dynamometer (T.K.K. 5401, Grip-D, Takei, Japan), adjusted by hand size for each participant. The participants were instructed to stand with their arms wholly extended, gradually and continuously squeezing the handgrip to the maximum of their strength, for at least 2 s. Participants performed the test twice with the dominant hand. A 90-s rest period was given between trials. The best score was recorded in kilograms [17].

### 2.5. Rate of Perceived Exertion (RPE)

Perception of effort was evaluated using RPE scores (10-point scale) [18] collected in all training sessions during the training period. Each participant’s RPE was collected 30 min after the end of the session to ensure that the rating reflected the whole training session and not only the last period [19]. 

### 2.6. Physical Activity Enjoyment Scale (PACES)

The original version of the PACES [20] included 18 items scored on a 7-point bipolar scale. The scale was intended to gauge the extent to which an individual enjoys doing any physical activity. The revised version consists of 16 statements scored on a 5-point Likert scale ranging from 1 (disagree a lot) to 5 (agree a lot). The stem for each item is “When I am active …”. Nine items are positive: “I enjoy it”, “I find it pleasurable”, “It gives me energy”, “It’s very pleasant”, “My body feels good”, “I get something out of it”, “It’s very exciting”, “It gives me a strong feeling of success”, “It feels good”. Seven items are negative: “I feel bored”, “I dislike it”, “It’s no fun at all”, “It makes me depressed”, “It frustrates me”, “It’s not at all interesting”, “I feel as though I would rather be doing something else”. High scores on the positive items and low scores on the negative items would indicate a high enjoyment of the physical activity. A total enjoyment score can also be obtained by reversing negative item scores and summing them to positive item scores. With this procedure, total enjoyment scores can range from 16 to 80 (maximum enjoyment).

### 2.7. Training Intervention

The recreational team volleyball training intervention was held for 10 weeks. The training was offered three times per week, and the participants were encouraged to participate in at least two sessions per week. During this period, the participants in the RV group performed 2/3 training sessions of 90 min per week, consisting of a standardized 10-min warm-up followed by 80 min of playing recreational team volleyball matches (4 vs. 4 and 3 vs. 3), interspersed by two 5-min breaks. The standardized warm-up consisted of 5 min of jogging, running at progressively increasing speeds, and 5 min of technical ball drills (passes). The training sessions were held with, at least, a 48-h rest period in between, in an indoor smaller volleyball court (16 × 8 m). Average total training attendance was 25 ± 7 sessions during the 10-week intervention period (i.e., 30 training sessions). The participants in the control group maintained their usual daily physical activity, and both groups reported no changes in their diet during the 10-week period.

### 2.8. Statistical Analysis

Descriptive statistics were calculated for age, height, and body mass. A repeated measure ANOVA (2 × 2) with Bonferroni post hoc multiple comparison tests was computed to test for interactions and main effects for time (pre-test vs. post-test) and group (RV vs. CON) on the selected physical test variables. Practical significance was assessed by calculating partial eta squared (η^2^*_p_*) (values of 0.01, 0.06, and above 0.15 were interpreted as small, medium, and large, respectively) [21]. All tests were carried out using SPSS, version 22 (SPSS Inc., Chicago, IL, USA), and assessed at the *p* ≤ 0.05 level of significance. The effect size (ES) for intragroup differences was classified as follows: <0.2 was defined as trivial; 0.2–0.6 was defined as small; 0.6–1.2 was defined as moderate; 1.2–2.0 was defined as large; 2.0–4.0 was defined as very large; and >4.0 was defined as extremely large [22].

## 3. Results

The average RPE after the intervention in the RV group was 3.14 ± 0.57 and 72 ± 5% HRmax, and the achieved score on the PACES enjoyment questionnaire was 70.91 ± 5.20.

Results from repeated measure ANOVA indicated a significant group (RV vs. CON) × time (pre to post) interaction for LDL (F = 6.776, *p* = 0.016, partial ƞ^2^ = 0.235; see Table 1).

When examining the impact of intervention on triglyceride concentration, there was a significant main effect for time (*p* = 0.008) with both groups improving their result after the 10 week intervention (RV − ES = −0.8, % change = −26.4% vs CON – ES = −0.3, % change = −8.8%). After 10 weeks, there was a significant main effect in blood glucose concentration for time (*p* < 0.001) with a 21.1% change in the RV group (ES = −1.86) and 18.5% change in the CON group (ES = −1.28).

Health marker results revealed a significant group (RV vs. CON) × time (pre to post) interaction for resting HR (F = 11.647, *p* = 0.002, partial ƞ^2^ = 0.346; see Table 2). There was a significant main effect for time (*p* = 0.039) for systolic BP. No significant changes were observed for body weight, BMI, and diastolic BP.

Results indicated a significant group (RV vs. CON) × time (pre to post) interaction for YYIRT2 (F = 11.648, *p* = 0.003, partial ƞ^2^ = 0.380; see Figure 1).

When examining the impact of the intervention on handgrip strength, there were no significant changes in both groups.

## 4. Discussion

To our knowledge, this is the first study examining the effect of small-sided recreational volleyball on health markers and physical fitness in middle-aged men. We found that 10 weeks of small-sided recreational volleyball decreased some risk factors, specifically, LDL, resting HR, and systolic BP, compared to the control group. Moreover, the RV group showed better results in cardiovascular fitness compared to the control group. These findings partially confirmed the study hypothesis with positive effects on cardiovascular fitness and some health markers.

### 4.1. Biochemical Markers

Recreational team sport interventions in the form of soccer, handball, and basketball played as small-sided games were reported to improve blood profile significantly [14,23,24]. A typical finding from the abovementioned studies was that total cholesterol and LDL cholesterol were lowered after a period of recreational training. The RV group reported a moderate post-training decrease in LDL (i.e., −7.3%) cholesterol. The reported practical changes in LDL values found in this study are in the middle range of those reported in recreational soccer and recreational handball studies that reported changes in LDL cholesterol (4%–15%) [14,23]. However, the changes in total cholesterol levels alongside the increase in HDL cholesterol achieved by the RV group were lower than those found in recreational soccer and handball [14,23]. The biggest impact of RV was reported for triglycerides (−26.4%) and glucose (−21.1%), although the control group also showed moderate to large improvements. The improvements in the control group were observed probably due to the fact that physical activity and diet were subjectively controlled and assessed via questionnaires. Blood glucose concentration is an indirect marker of insulin resistance that can be affected by aerobic and resistance exercise [25,26]. The RV group showed a large (21%) decrement in fasting blood glucose, a finding different from that reported in recreational handball, soccer, and street basketball training interventions [14,23,24]. This change in blood glucose is of great importance, given the constant quest for new exercise strategies to prevent a type II diabetes pandemic. Therefore, further studies are warranted.

### 4.2. Health Markers

Resting HR has been suggested as a non-invasive, powerful, and independent predictor of cardiovascular diseases [27,28]. The increase in resting HR above at least 60 bpm could lead to increased risk of the mentioned diseases [29]. The RV group reported a moderate, 4.2%, post-intervention decrement in resting HR. The raw change (−3 bpm) in resting HR of RV practice was lower than that reported by Póvoas et al. [14] in recreational handball (−10 bpm) but also lower than values found by Randers et al. [24] (−7 bpm) in healthy individuals, Schmidt et al. [30] in untrained elderly, and Andersen et al. [31] in mild hypertensive individuals (i.e., −8 bpm) participating in recreational soccer interventions. Our results are comparable to those found in full-court (4 bpm) and half-court (2 bpm) recreational basketball [24], and full-court recreational volleyball (5 bpm) [8] in middle-aged men. However, baseline values for recreational basketball players in the study of Randers et al. [24] were very low (~59 bpm), so a further drop was not expected. Therefore, recreational volleyball played on a full-size or smaller court elicits smaller changes compared to the other above-mentioned recreational team-based activities.

Several studies have shown that team-sport activities such as recreational football, volleyball, basketball, and handball effectively decrease blood pressure [8,14,24,32]. Moreover, one study revealed that only eight sessions of high-intensity interval training and moderate-intensity continuous training programs improved systolic blood pressure [33]. In this study, systolic blood pressure was lowered after training in the RV group with no change in the CON group. The RV group experienced a 2.4%, and 1.2% post-intervention decrement in systolic blood pressure and diastolic blood pressure, respectively. This finding indicates that small-sided volleyball recreational practice positively affected the cardiovascular system of the participants. The 3 mmHg reduction in SBP was lower than the drop reported in untrained participants that volunteered in soccer, handball, and street basketball training studies (4–8 mmHg) [14,23,24], but also lower than values found in whole court recreational volleyball (4 mmHg) [8].

Obesity can be considered as a risk factor for several chronic diseases [24]. The participants in the present study had baseline BMI values just above 25 and can thus be considered overweight, so lowering body weight would be important for an improved health profile. After the training period, change in total body mass was low (0.6%), which is similar to results found after whole and half-court recreational basketball, and after small-sided football training [32].

Recent reports stated that vigorous physical activity seems to provide a number of health benefits and promote healthy ageing [34,35,36]. Additionally, Poček et al. [37] concluded in their review that health benefits of physical activities depend on engagement at recommended levels. Therefore, to achieve better effects on health it was suggested that exercise should be vigorous. However, recreational volleyball is not as intensive as professional volleyball and accordingly not as effective in improving health markers.

### 4.3. Physical Fitness

The 10 weeks of small-sided recreational volleyball practice produced only a 2.4% change (8 m) in players’ YYIRT2 performance with no change in handgrip strength (0.5%). Volleyball represents an explosive and fast-paced activity [38]. Therefore, it was expected that changes in strength in our study would be somewhat greater. However, it seems that external and internal load in recreational volleyball is notably lower than in professional volleyball. The achieved YYIRT2 performance change was significantly lower than those reported in recreational handball (80%) [14] and recreational soccer studies (37% to 49%) [39] after similar intervention durations and addressing different populations. Smaller improvements in recreational volleyball compared to recreational handball and soccer could be due to different tests used in handball (Yo-yo intermittent endurance test level-2) and soccer (YYIRT1). In a similar study [8] conducted on the whole court, the recreational volleyball group improved their VO_2_ max performance by 4.3% between pre- and post-tests, while a 3.2% decrease was observed in the control group. This change is in line with our results, although the mentioned study used a shuttle run test to determine cardiovascular fitness. Smaller improvements in cardiovascular fitness following recreational volleyball training compared to soccer, handball, and basketball were mainly due to lower intensity in RV (72% HRmax and 3.14 RPE). Milanovic et al. [39] stated that the intensity of 78–84% maximal heart rate (HRmax) during recreational soccer is enough for healthy untrained men to improve their VO^2^ max by 8–13%. Therefore, it is likely that high-intensity periods make recreational soccer, basketball, and handball training superior to volleyball training in terms of producing improvements in VO_2_ max. Moreover, we could speculate that greater improvements may occur with different warm-up protocols. This was confirmed with one study, albeit in young adults, that showed that six weeks of dynamic and static stretching performed three times per week had a positive effect on sprint performance in recreational male volleyball players [40]. However, the strength of this study could be the fact that small-sided recreational volleyball training has been shown to be a promising training protocol for increasing exercise enjoyment and promoting exercise adherence in sedentary adults aged 35–55 years.

A study limitation may be the lack of objectively measured physical activity and nutritional control in both RV and CON. Participants were asked to continue their usual diet and to avoid any other physical activity programs. This might have affected the training effect on some health markers. The small sample size is also considered as a limitation. Moreover, the participants in the RV group had previous experience with volleyball training. Therefore, interventions aimed at analyzing the effects of recreational volleyball practice on participants of either gender with little or no experience in this sport are warranted.

## 5. Conclusions

Recreational volleyball played with fewer players on a smaller court can be an effective training modality to stimulate decrease in LDL cholesterol and resting HR. However, it was not beneficial in improving physical fitness, with a small improvement in cardiovascular fitness only. Nevertheless, this study shows that only two recreational volleyball sessions per week can give meaningful benefits compared to the control group. This is of great importance considering that people with limited leisure time can practice recreational volleyball two times per week and still gain health benefits.

## Figures and Tables

**Figure 1 ijerph-17-03021-f001:**
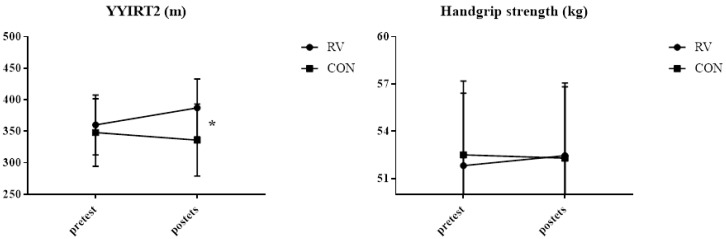
Physical fitness pre-post results. Abbreviations: YYIRT2, Yo-Yo intermittent recovery test–level 2; RV, recreational volleyball group; CON, control group; *, significant interaction between group and time *p* < 0.05.

**Table 1 ijerph-17-03021-t001:** Biochemical markers.

Group		Pretest	Posttest	ES	% Change	*p*-Value, η^2^*_p_*
Total cholesterol (mmol/L)						
	RV	3.58 ± 0.41	3.59 ± 0.43	+0.02	+0.3%	Group: *p* = 0.031, η^2^*_p_*: 0.12 Time: *p* = 0.628, η^2^*_p_*: 0.011 Interaction: *p* = 0.173, η^2^*_p_*: 0.083
	CON	4.0 ± 0.97	4.05 ± 1.06	+0.05	+1.3%	
HDL (mmol/L)						
	RV	1.28 ± 0.24	1.32 ± 0.24	+0.17	+3.1%	Group: *p* = 0.572, η^2^*_p_*: 0.015 Time: *p* = 0.410, η^2^*_p_*: 0.031 Interaction: *p* = 0.810, η^2^*_p_*: 0.003
	CON	1.22 ± 0.24	1.28 ± 0.27	+0.23	+4.9%	
Triglycerides (mmol/L)						
	RV	1.25 ± 0.46	0.92 ± 0.34	−0.82	−26.4%	Group: *p* = 0.058, η^2^*_p_*: 0.154 Time: *p* = 0.008, η^2^*_p_*: 0.278 Interaction: *p* = 0.208, η^2^*_p_*: 0.071
	CON	1.47 ± 0.43	1.34 ± 0.49	−0.28	−8.8%	
Glucose (mmol/L)						
	RV	5.25 ± 0.56	4.14 ± 0.63	−1.86	−21.1%	Group: *p* = 0.838, η^2^*_p_*: 0.002 Time: *p* < 0.001, η^2^*_p_*: 0.580 Interaction: *p* = 0.693, η^2^*_p_*: 0.007
	CON	5.13 ± 0.83	4.18 ± 0.64	−1.28	−18.5%	
LDL (mmol/L)						
	RV	2.73 ± 0.31	2.53 ± 0.38	−0.58	−7.3%	Group: *p* = 0.813, η^2^*_p_*: 0.003 Time: *p* < 0.001, η^2^*_p_*: 0.493 Interaction: *p* = 0.016, η^2^*_p_*: 0.235
	CON	2.63 ± 0.35	2.57 ± 0.34	−0.17	−2.3%	

Abbreviations: RV, recreational volleyball group; CON, control group; HDL, high-density lipoprotein; LDL, low-density lipoprotein; ES, effect size.

**Table 2 ijerph-17-03021-t002:** Health markers.

Group		Pretest	Posttest	ES	% Change	*p*-Value, η^2^*_p_*
Body weight (kg)						
	RV	85.42 ± 6.91	84.92 ± 6.43	−0.07	−0.6%	Group: *p* = 0.334, η^2^*_p_*: 0.042 Time: *p* = 0.760, η^2^*_p_*: 0.004 Interaction: *p* = 0.228, η^2^*_p_*: 0.065
	CON	87.00 ± 4.31	87.83 ± 4.90	+0.18	+1.0%	
Rest HR (bpm)						
	RV	67.67 ± 3.65	64.83 ± 3.16	−0.83	−4.2%	Group: *p* = 0.135, η^2^*_p_*: 0.09 Time: *p* < 0.001, η^2^*_p_*: 0.640 Interaction: *p* = 0.002, η^2^*_p_*: 0.346
	CON	68.50 ± 2.71	67.67 ± 2.23	−0.33	−1.2%	
BMI (kg/m^2^)						
	RV	25.85 ± 1.74	25.72 ± 1.76	−0.07	−0.5%	Group: *p* = 0.958, η^2^*_p_*: 0.000 Time: *p* = 0.674, η^2^*_p_*: 0.008 Interaction: *p* = 0.215, η^2^*_p_*: 0.069
	CON	25.62 ± 1.48	25.88 ± 1.39	+0.18	+1.0%	
Systolic BP (mm Hg)							
	RV	132.6 ± 10.4	129.4 ± 8.2	−0.34	−2.4%	Group: *p* = 0.599, η^2^*_p_*: 0.013 Time: *p* = 0.039, η^2^*_p_*: 0.179 Interaction: *p* = 0.061, η^2^*_p_*: 0.150
	CON	132.9 ± 8.5	132.8 ± 7.1	−0.01	−0.1%	
Diastolic BP (mm Hg)							
	RV	86.42 ± 4.34	85.42 ± 4.52	−0.23	−1.2%	Group: *p* = 0.540, η^2^*_p_*: 0.017 Time: *p* = 0.114, η^2^*_p_*: 0.109 Interaction: *p* = 0.334, η^2^*_p_*: 0.042
	CON	87.08 ± 4.38	86.83 ± 3.49	−0.06	−0.3%	

Abbreviations: RV, recreational volleyball group; CON, control group; HR, heart rate; BP, blood pressure; BMI, body mass index; ES, effect size.

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
