# Peer review of "Effects of Small-Sided Recreational Volleyball on Health Markers and Physical Fitness in Middle-Aged Men"

_ijerph, 2020, doi:10.3390/ijerph17093021_

Round 1

Reviewer 1 Report

The Authors stated (line 66) The subjects were matched and randomly assigned to a small-sided recreational volleyball 66 group (RV; n = 14; age: 44.7 ± 6.34 years; height: 181.8 ± 6.1 cm), or a control group - performing no 67 organized physical training (CG; n = 12; age: 42.9 ± 8.72 years; height: 184.4 ± 5.4 cm). Thus, only volleyball was the sport exercise submitted to 26 subjects, but at line 145 there are different sport disciplines. Moreover, it sis not explained the distribution of subjects among different disciplines.

Diet is not recorded, then the evaluation of  lipids (HDL, cholesterol, triglycerides) and glucose is unuseful

Reviewer 2 Report

1.- line 37: describe the type of barriers.
2.- How was the control group determined?
3.- In the limitations, the reduced sample size must be indicated.
3.- Why did they not perform other tests to evaluate the strength in the lower limbs?

Reviewer 3 Report

General comment

This is a well-written paper on an interesting topic that has not been studied sufficiently in the existed literature. Considering the popularity of recreational volleyball, I would recommend it for publication once the author addressed a few concerns.

Specific comments

  1. Merge small paragraphs into larger
  2. Abstract: Add details on participants’ demographic characteristics.
  3. Abstract: Add the direction of interactions and effect sizes.
  4. Abstract: Be precise on the conclusions; report the areas of improvement.
  5. Use convenient abbreviations. Change RV to EXP (experimental group).
  6. Abstract: Add also a conclusion with broader perspective.
  7. Introduction: Introduce all the variables presented in methods.
  8. 59: Add the rationale for the study. Why is it important to study this topic?
  9. 61: Add hypotheses after aims.
  10. 105: What is RPE?
  11. 106: Add details about the assessment of RPE.
  12. 108: What is PACES?
  13. Discussion: The authors should not simply describe their data with regards to the literature, but also they should go deeper in the interpretation of the findings.
  14. References: Relevant literature from IJERPH -that could enhance the discussion- is missing (PMID: 31398904/ PMID: 31426481)

Round 2

Reviewer 1 Report

No comment

Reviewer 3 Report

Thank you for addressing my comments.

Please, check some minor typos in the parts you added in the revised version and the reference list.